# Deferred Administration of Afobazole Induces Sigma1R-Dependent Restoration of Striatal Dopamine Content in a Mouse Model of Parkinson’s Disease

**DOI:** 10.3390/ijms21207620

**Published:** 2020-10-15

**Authors:** Ilya A. Kadnikov, Ekaterina R. Verbovaya, Dmitry N. Voronkov, Mikhail V. Voronin, Sergei B. Seredenin

**Affiliations:** 1Department of Pharmacogenetics, FSBI “Zakusov Institute of Pharmacology”, Baltiyskaya street 8, 125315 Moscow, Russia; zueva.kate1997@gmail.com (E.R.V.); dnamed@mail.ru (M.V.V.); 2Laboratory of neuromorphology, Research Center of Neurology, Volokolamskoe Highway 80, 125367 Moscow, Russia; voronkovdm@gmail.com

**Keywords:** chaperone Sigma1R, Parkinson’s disease, 6-OHDA, dopamine, rotarod, tyrosine hydroxylase, afobazole, PRE-084, BD-1047

## Abstract

Previously, we demonstrated that the immediate administration of multitarget anxiolytic afobazole slows down the progression of neuronal damage in a 6-hydroxidodamine (6-OHDA) model of Parkinson’s disease due to the activation of chaperone Sigma1R. The aim of the present study is to evaluate the therapeutic potential of deferred afobazole administration in this model. Male ICR mice received a unilateral 6-OHDA lesion of the striatum. Fourteen days after the surgery, mice were treated with afobazole, selective Sigma1R agonist PRE-084, selective Sigma1R antagonist BD-1047, and a combination of BD-1047 with afobazole or PRE-084 for another 14 days. The deferred administration of afobazole restored the intrastriatal dopamine content in the 6-OHDA-lesioned striatum and facilitated motor behavior in rotarod tests. The action of afobazole accorded with the effect of Sigma1R selective agonist PRE-084 and was blocked by Sigma1R selective antagonist BD-1047. The present study illustrates the Sigma1R-dependent effects of afobazole in a 6-OHDA model of Parkinson’s disease and reveals the therapeutic potential of Sigma1R agonists in treatment of the condition.

## 1. Introduction

Parkinson’s disease (PD) is the second most common chronic progressive neurodegenerative disorder of insidious onset, which is characterized by the presence of predominantly motor symptomatology: bradykinesia, rest tremor, rigidity, and postural disturbances [1,2]. Motor impairment is combined with nonmotor symptoms such as depression, anxiety, sleep abnormalities, constipation, and cognitive decline [3]. The prevalence of the disease rises with age [4].

Two forms of PD are defined: familial and idiopathic. At present, the idiopathic form of PD, which is most prevalent, is considered to be a result of gene‒environment interaction. The most common factors that may cause PD are pesticides, heavy metals, and drug abuse [5].

Histologically, PD is characterized by neuronal loss in the dopamine (DA)-producing region of the substantia nigra (SNc) and widespread intracellular accumulation of α-synuclein within Lewy bodies (LBs) [6]. In the prodromal period before the manifestation of severe symptoms of the disease [7], several interrelated processes underlying neuronal damage take place: the activation of oxidative stress [8], mitochondrial dysfunction [9], ER stress [10], α-synuclein misfolding and aggregation [11,12], dysfunction in protein clearance systems [13,14,15], and neuroinflammation [16].

The only treatment option for PD is symptomatic, i.e., compensating for DA loss. The most commonly used medication is levodopa (DOPA), in combination with peripheral DOPA decarboxylase inhibitors, agonists of DA receptors, inhibitors of monoamine oxidase B (MAO-B) and catechol O-methyltransferase (COMT), β-blockers, and anticholinergics [17]. The shortcomings of this therapy are the development of medication-related complications, potential side effects, and diminished effectiveness over time [18,19].

One of the promising targets for PD treatment is chaperone Sigma1R [20,21,22]. Sigma1Rs are widely spread in the brain and nigrostriatal pathway in particular [23,24]. In patients with early-stage PD, striatal damage is accompanied by a decrease in the Sigma1R level [25], which corresponds to the high death rate of dopaminergic neurons in *Sigmar1*^−/−^ mice [23]. Intracellular Sigma1Rs are predominantly located within the cholesterol reach region of the endoplasmic reticulum (ER) membrane in contact with mitochondrion (MAM) [26]. The ligand activation of Sigma1R provides its translocation [27] and chaperone activity toward multiple proteins of plasmatic, ER, and nucleus membranes [28]. Those are D_1_ and D_2_ dopamine receptors [29], dopamine transporter (DAT) [30], GluNs [31,32], neurotrophine receptor trkB [33], IP_3_R3 within the MAM region [34], main ER chaperone BiP [34,35], ER stress sensor IRE1 [36], and nucleus membrane protein emerin [37]. These interactions modulate the activity of client proteins [29,30,31,32,33,34,37,38,39,40] and may contribute to the protective effect in dopaminergic neurons of nigrostriatal tracts [10,41,42,43,44,45,46,47,48].

A growing body of evidence allows considering PD not only as a protein-misfolding disease [49], but also as lipidopathy [50,51,52]. This hypothesis is based on the interrelation of the protein folding and lipid metabolisms that triggers neuronal damage and LBs formation [50,51]. Therefore, the pharmacological regulation of Sigma1R may have a neuroprotective effect in PD. The ability of Sigma1R to form cholesterin-containing lipid domains and participate in compartmentalization and lipid transfer from ER to lipid droplets or the region of plasmatic membrane is widely known [53,54]. Ligand activation promotes the formation of a chaperone Sigma1R complex with insulin-induced gene 1 protein (INSIG1), which is a member of the ER-associated degradation system (ERAD) that regulates lipids and cholesterol homeostasis by acting on the degradation of lipid-synthesizing enzymes [28,55].

Sigma1R activation contributes to the diminishing of oxidative stress [56]—one of the factors of PD pathogenesis [8]. Recent studies demonstrate the ability of Sigma1R agonists to upregulate complex I activity (NADH ubiquinone oxidoreductase) and reduce reactive oxygen species (ROS) production caused by Aβ_1–42_ in mouse forebrain mitochondria [57]. In addition, the role of Sigma1R in the regulation of neuroinflammation was specified [58].

One of the common approaches in PD modeling in rodents is a unilateral intrastriatal injection of 6-OHDA [59,60]. This neurotoxin causes the death of dopaminergic neurons of SNc by ROS activation [61], an inhibition of mitochondrial complex I [61], microglia activation [62], neuroinflammation [63], and autophagy dysfunction [64]. The lack of LBs in neurons is considered to be a shortcoming of the 6-OHDA-based model [60]. However, heterogeneous data on the nature of LBs, absence of uniformity in morphology, and biochemical composition of LB-like inclusions in different models of PD, as well as the reproducibility of LBs properties specific for PD patients, make this limitation of the 6-OHDA model debatable [65,66]. Nevertheless, there is evidence that α-synuclein is involved in the lesions caused by 6-OHDA [67]. 6-OHDA increased, in a concentration-dependent manner, the content of monomeric and oligomeric α-synuclein in SH-SY5Y cells in vitro and enhanced the phosphorylation of α-synuclein (pSyn-129) characteristic for PD [68], which is considered to be an LB-like feature [65]. These properties of the 6-OHDA model are consistent with the changes in the brains of *Sigmar1* knockout mice [23]. The biphasic time course of neurodegeneration [69], weak progression of nigrostriatal tract damage, and absence of LBs caused by a single injection of the toxin at the prerequisite dose [70,71] correspond to the early stage of PD [6].

Different in vivo models of PD have demonstrated the possibility of neuroprotective activity due to Sigma1R ligand activation [20,22,72]. The selective Sigma1R agonist PRE-084, administered to mice with unilateral intrastriatal 6-OHDA lesions, restored DA content in the striatum, increased the number of TH+ cells in SNc, and facilitated motor behavior in various tests [20,22]. The Sigma1R agonist with antidepressant activity, SA4503, improved both motor deficits and cognitive impairments in 1-methyl-4-phenyl-3,6-dihydro-2H-pyridine (MPTP)-treated mice [72].

The anxiolytic drug afobazole (5-ethoxy-2-[2-(morpholino)-ethylthio]benzimidazole dihydrochloride) was developed and pharmacologically studied at the FSBI Research State Zakusov Institute of Pharmacology [73]. Afobazole has an affinity to chaperone Sigma1R (Ki = 5.9 µM) and regulatory sites of ribosyldihydronicotinamide dehydrogenase [quinone] (NQO2, Ki = 0.97 µM) and monoamine oxidase A (MAO-A, Ki = 3.6 µM) [74,75]. In vitro and in vivo experiments suggest a role of Sigma1R in the mechanisms of previously described cytoprotective [76] and neuroprotective [77,78,79,80,81,82] properties of afobazole. Afobazole restored DA content in the striatum of mice with unilateral intrastriatal 6-OHDA lesions [83]. Previously, in a 6-OHDA-induced model of PD using selective Sigma1R ligands, we demonstrated the contribution of chaperone Sigma1R to the neuroprotective effect of a 14-day afobazole course started on the day of 6-OHDA lesion [22]. The aim of the present investigation became the study of Sigma1R’s contribution to the action of afobazole upon deferred administration to mice with intrastriatal 6-OHDA lesions.

## 2. Results

In a neurochemical study, we have observed the significant effect (in a two-way ANOVA test) of 6-OHDA injection on dopamine content in the striatum (F = 69.31; *p* < 0.0001). The action of the tested compounds on the DA content in the striatum of sham-operated and 6-OHDA-lesioned mice was different (F = 4.716; *p* = 0.0007). The interaction between 6-OHDA injection and treatment had a statistically significant effect (F = 11.88; *p* < 0.0001).

The DA content in the contralateral striata of mice with 6-OHDA lesions did not vary from that of sham-operated animals when the DA content in respective ipsilateral striata was significantly lower (Figure 1, Appendix A). Dopamine content in the striatum with 6-OHDA lesions of vehicle-treated mice was almost half that in the contralateral striatum and damaged striatum of vehicle-treated sham-operated mice (Figure 1). None of the tested compounds affected dopamine content in the intact or damaged striatum of sham-operated animals (Figure 1). Afobazole administered at a dose of 2.5 mg/kg over 14 days with a course start 14 days after surgery facilitated the DA content in the 6-OHDA-lesioned striatum, bringing it to the level of the damaged striatum of sham-operated animals treated with the vehicle (Figure 1). The administration of Sigma1R agonist PRE-084 at a dose of 1.0 mg/kg had the same effect on the DA content (Figure 1). Sigma1R antagonist BD-1047 treatment at a dose of 3.0 mg/kg had no impact on 6-OHDA lesions, whereas its daily administration 30 min prior to afobazole or PRE-084 blocked the effects of both drugs on DA content (Figure 1).

Changes in DA metabolites content dihydroxyphenylacetic acid (DOPAC) and homovanillic acid (HVA) fit the same pattern as DA. Both 6-OHDA injection (F = 65.53; *p* < 0.0001 for DOPAC and F = 40.23; *p* < 0.0001 for HVA) and the compounds used (F = 5.71; *p* = 0.0001 for DOPAC and F = 3.62; *p* = 0.0049 for HVA) had an effect on their content. The interaction between 6-OHDA injection and treatment had a statistically significant effect (F = 7.64; *p* < 0.0001 for DOPAC and F = 11.02; *p* < 0.0001 for HVA).

In accordance with the DA content, the levels of its metabolites DOPAC and HVA in the 6-OHDA-lesioned striatum of vehicle-treated mice were also decreased compared to those of the contralateral striatum and damaged striatum of vehicle-treated sham-operated mice (Figure 2 and Figure 3). Both the administration of afobazole and Sigma1R agonist PRE-084 increased DOPAC and HVA content to the level of relevant families of comparison (Figure 2 and Figure 3). The administration of Sigma1R antagonist BD-1047 had no effect on DOPAC and HVA content. Pre-administration of BD-1047 significantly prevented the effects of afobazole and PRE-084 (Figure 2 and Figure 3).

Metabolic ratios of DOPAC/DA and (DOPAC+HVA)/DA were increased in the ipsilateral striatum of 6-OHDA-lesioned mice, whereas afobazole and PRE-084 increased HVA/DA and (DOPAC+HVA)/DA ratios (Appendix A).

In the immunohistochemical study, the number of tyrosine hydroxylase-positive (TH+) cells in the ipsilateral SNc of vehicle-treated mice with 6-OHDA lesions was decreased by 35% compared to that of sham-operated animals and by 38% as compared to the contralateral side (Table 1). Despite the deferred chronical treatment with afobazole increasing the DA content in 6-OHDA-lesioned striatum, the number of TH+ cells in SNc of those animals was almost equal to that of vehicle-treated mice with 6-OHDA lesions (Table 1). The difference in the relative amount of TH+ cells in the SNc of afobazole-treated animals was insignificant compared to the amount observed in 6-OHDA-lesioned mice (Table 1).

In fixed-speed (FSRR) and accelerated (ARR) variations of the rotarod test, the median latency to fall of vehicle-treated mice with 6-OHDA lesions significantly decreased compared to that of sham-operated animals (Figure 4 and Figure 5, Appendix A). Selective Sigma1R agonist PRE-084 and afobazole managed to restore the ability to hold on to the rod of animals with 6-OHDA lesions (Figure 4 and Figure 5). Administration of BD-1047 had no effect on the latency to fall of hemiparkinsonian animals. In both variations of the test, the pre-administration of BD-1047 markedly prevented the action of PRE-084 and afobazole (Figure 4 and Figure 5). All tested compounds and their combinations had no effect on the motor behavior of sham-operated animals (Figure 4 and Figure 5). Our data show a moderate positive correlation between DA content in the ipsilateral striatum and latency to fall in FSRR (R = 0.6, *p* < 0.0001) and ARR (R = 0.6, *p* < 0.0001) tests (Figure 6 and Figure 7).

## 3. Materials and Methods

### 3.1. Chemicals

The following chemicals were used: afobazole (5-ethoxy-2-[2-(morpholino)-ethylthio]benzimidazole dihydrochloride) (FSBI Research State Zakusov Institute of Pharmacology, Moscow, Russia), PRE-084, BD-1047 (Tocris Bioscience, Bristol, UK), 6-hydroxydopamine hydrochloride (6-OHDA), ascorbic acid, NaCl, sucrose, paraformaldehyde (PFA), polyclonal antibodies against tyrosine hydroxylase T8700, secondary antibodies conjugated with CF488 fluorochrome SAB4600045, FluoroShield, 4′,6-diamidino-2-phenylindole, Triton X-100, chloral hydrate, 3,4-dihydroxybenzylamine hydrobromide (DHBA), dopamine (DA), 3,4-dihydroxyphenylacetic acid (DOPAC), homovanillic acid (HVA), KH_2_PO_4_, H_3_PO_4_, HClO_4_, citric acid, ethylenediaminetetraacetic acid disodium salt dehydrate (EDTA-Na_2_), octanesulfonic acid, acetonitrile, phosphate-buffered saline (PBS) (Sigma-Aldrich, St. Louis, MO, USA), and Tissue Tek O.C.T. medium (Sakura, Osaka, Japan).

### 3.2. Experimental Animals

The study was performed in male ICR (CD-1) mice (25–30 g, *n* = 105) obtained from Pushchino Breeding Center (Branch of the Institute of Bioorganic Chemistry, Russian Academy of Sciences). Animals were housed under standard vivarium conditions (20–22 °C, 30–70% humidity, 12-h light/dark cycle) in plastic cages with sawdust bedding and 6‒12 animals per cage.

### 3.3. Ethical Approval

All experimental procedures were approved by the bioethics committee of the FSBI Research Zakusov Institute of Pharmacology, protocol #01‒1 of 27 January 2020. All applicable national [84] and international [85] guidelines for the care and use of experimental animals were followed.

### 3.4. Experimental Design

The experimental design was developed in compliance with the 3R principles. All animals received unilateral intrastriatal injection of 6-OHDA or vehicle solution (sham-operated mice). Fourteen days after the surgery, daily administration of vehicle, afobazole, PRE-084, BD-1047, or a combination of BD-1047 with afobazole or PRE-084 was started and ran for another 14 days. All drug substances were dissolved in water for injections immediately before administration and were intraperitoneally injected at a volume of 0.1 mL/10 g body weight. The tested compounds were administered at the following doses: afobazole—2.5 mg/kg, selective Sigma1R agonist PRE-084—1.0 mg/kg, selective Sigma1R antagonist BD-1047—3.0 mg/kg, in case of combined administration BD-1047 was injected 30 min prior to afobazole or PRE-084. On the 28th day, all animals passed the rotarod test and were vivisected for brain tissue samples for DA content measurement. Among these animals, vehicle-treated sham-operated, and vehicle- and afobazole-treated 6-OHDA-lesioned mice were picked for immunohistochemical assay using the random number calculator, five animals per group. The animals were divided as follows. Mice with 6-OHDA lesions: vehicle-treated mice (*n* = 12), mice treated with afobazole (*n* = 12), PRE-084 (*n* = 9), BD-1047 (*n* = 10), and BD-1047 following the administration of afobazole (*n* = 11) or PRE-084 (*n* = 10). Sham-operated mice received vehicle (*n* = 10), afobazole (*n* = 6), PRE-084 (*n* = 6), BD-1047 (*n* = 6), or a combination of BD-1047 and afobazole (*n* = 6) or PRE-084 (*n* = 6). To calculate the sample sizes, a power analysis was used.

### 3.5. 6-OHDA Lesion

6-OHDA lesions were made according to [22]. In detail, 30 min prior to surgery, the animals were anesthetized with chloral hydrate (400 mg/kg intraperitoneally) [67,86]. Anesthetized animals were placed into the stereotaxic frame (Stoelting Co., Wood Dale, IL, USA), and 6-OHDA was injected into the right striatum according to the coordinates A = 0.4, L = 1.8, and V = −3.5 relative to bregma [67]. 6-OHDA was dissolved at 5 μg per 1 μL of a solution containing 0.9% NaCl and 0.02% ascorbic acid. The experimental animals were injected with 1 μL of the 6-OHDA solution at a rate of 0.5 μL/min using a Hamilton syringe equipped with a 30-gauge stainless steel needle. The needle was withdrawn 2 min after the injection. Sham-operated animals were injected with 1 μL of saline with 0.02% ascorbic acid at the same coordinates.

### 3.6. Rotarod Test

Motor behavior was studied in male ICR mice utilizing a Rota-rod/RS LE 8500 apparatus (Panlab/Harvard apparatus, Barcelona, Spain). The rotarod test was performed as described previously [22,87]. Two training sessions spaced 24 h apart were performed to acclimate the experimental animals to the apparatus and to exclude hypodynamic animals from the study. The first training session was performed on the 26th day after surgery. Each animal was placed on the rod twice as it rotated at a rate of 4 rpm, with a 1 h interval between sessions. During the second session, the rotation speed was increased to 10 rpm. Animals unable to hold on to the rod for more than 1 min were excluded from the study [88]. In the present study, all animals were able to hold on to the rod for more than 1 min, so none of them were excluded from the experiment.

On the 28th day after surgery, rotarod testing was performed. For a more detailed analysis of the influence of Sigma1R ligands on the mice motor behavior fixed-speed rotarod (FSRR) and accelerating rotarod (ARR) tests were utilized. For the FSRR variation of the test, mice were placed onto a rod rotating at 20 rpm, and the latency to fall was measured. In the ARR version of the test, the experimental animal was placed on the rod, rotating at an initial speed of 4 rpm, and the time before falling onto the platform was measured. The maximum speed (40 rpm) was reached in 5 min. Each animal passed both versions of the test three times, with 30-min intervals between attempts. The time measurement stopped after animals stayed on the rod for 120 s with a fixed speed of rotation and 180 s for the increasing rate. The maximum time of the three attempts was used for statistical analysis. On days with training sessions and the rotarod test, drugs were administered after pretraining or testing. In terms of rotarod test, the last drugs administration was performed on the 27th day after the surgery 20 h prior to testing.

### 3.7. HPLC-ED Technique

Dopamine and its metabolites were measured in the damaged and intact striata of the mouse brain using the high-performance liquid chromatography technique with electrochemical detection (HPLC-ED), as described previously [22]. On the twenty-eighth day after the 6-OHDA administration, 1-h post drugs administration, the mice were decapitated, and the brains were removed. The left and right striata were dissected on wet ice covered with filter paper dampened in 0.32 M sucrose solution at a temperature of 0–4 °C. Each striatum was frozen in liquid nitrogen (‒196 °C), weighed, and stored at −80 °C. To measure the contents of dopamine and its metabolites, the striatum was homogenized in 0.1 M HClO_4_ in a TissueLyserLT bead homogenizer (Qiagen, Hilden, Germany) at a frequency of 45 bit/min for 5 min. DHBA was added as an internal standard at a concentration of 0.25 nmol/mL. The samples were centrifuged at 10,000× g and 4 °C for 10 min. Twenty microliters of the supernatant were applied to a Kromasil C-18 4.6 × 150 analytical column (Dr. Maisch, Ammerbuch, Germany) using an SIL-20 ACHT autosampler (Shimadzu, Kyoto, Japan). Monoamines and their metabolites were separated in the column using a 0.1 M citrate–phosphate buffer containing 0.3 mm sodium octanesulfonate, 0.1 mm EDTA-Na_2_, and 8% acetonitrile (pH 3.0) as the mobile phase. The determination of dopamine and its metabolites was performed using a Shimadzu LC-20 Prominence chromatographic station equipped with an ESA 5011 (E1 = −175; E2 = +250) electrochemical cell and a Coulochem III electrochemical detector (ESA, Chelmsford, MA, USA). The flow rate of the mobile phase was 1 mL/min. The results were processed on a PC using Multichrom 1.5 software (Ampersand, Moscow, Russia). MAO and catechol-O-methyltransferase (COMT) are involved in the biodegradation of dopamine. Therefore, we used the metabolic ratio DOPAC/DA, which reflects MAO activity; HVA/DA, which reflects COMT activity; and the total ratio (DOPAC + HVA)/DA [89].

### 3.8. Immunohistochemical Analysis

The immunohistochemical analysis was performed as described in [22] with slight modifications. Twenty-eight days after the surgery, the mice were decapitated, and the brains were extracted. Specimens were fixed for 24 h by immersion in 4% PFA solution, soaked in 30% sucrose, and embedded in Tissue Teck O.S.T. medium. Twelve-micrometer-thick serial frontal sections of the midbrain in the region of the SNc were prepared utilizing a Sakura Tissue Tec Cryo3 cryotome. Prior to staining, heat epitope retrieval was performed in a 0.1 M citric buffer solution (pH 6.0) in a microwave oven for repeated cycles (600 W, 5 min). A large water bath that contained vials with slides and a mercury thermometer was placed into a microwave oven. Heating was intermittent, so the temperature stayed at 93–96 °C. After cooling, the slides were rinsed with PBS and left for 1 h in PBS containing 0.1% Triton X-100. To reveal dopamine-producing neurons in the SNc, slides were incubated with rabbit polyclonal antibodies against tyrosine hydroxylase (1:500 dilution) for 16 h. Bound rabbit immunoglobulins were visualized by goat antibodies conjugated with CF488 fluorochrome. Primary and secondary antibodies were incubated at room temperature in a light-proof dampening chamber. Nuclei were counterstained with DAPI. All slides were coverslipped in a Fluoroshield mounting medium. Imaging was carried out using a Cytation 5 imaging reader (BioTek Instruments Inc., Winooski, VT, USA) equipped with 4x Olympus objective and DAPI and GFP filter cubes.

The count of TH+ cells was performed in images acquired from 3–6 15-µm-thick frozen frontal slices taken with a 75-µm gap on the −3.0 to −3.7 mm level with respect to bregma under the same microscopy conditions, i.e., exposition time 800 ms, gain 1.1. On the panoramic image, the pars compacta region was manually marked out, and all neuron bodies with a visible nucleus were marked using ImageJ software (National Institutes of Health, Bethesda, MD, USA).

### 3.9. Statistical Analysis

To evaluate the experimental data distribution, D’Agostino‒Pearson and Shapiro‒Wilk tests were used. Statistical significance was calculated using two-way ANOVA with Tukey’s post hoc test or Kruskal‒Wallis test with Dunn’s post hoc test. A paired *t*-test was used for dependent variables. Data are presented as the mean and standard deviation (mean ± S.D.) or median with lower and upper quartiles (Mdn (q25–75)). Linear coupling was performed using Spearman correlation analysis. A value of *p* < 0.05 was considered to be statistically significant. Statistical analysis and visualization were performed using GraphPad Prism software version 8.0.1 for Windows (GraphPad, La Jolla, CA, USA, www.graphpad.com).

## 4. Discussion

In the present study, a 14-day course of afobazole at a dose of 2.5 mg /kg i.p., started two weeks after unilateral intrastriatal injection of 6-OHDA, caused an increase in the DA striatal content and latency to fall in two variations of a rotarod test. Intrastriatal DA decrease is a marker of nigrastriatal tract damage [90]. The restoration of DA in response to the therapy within the frameworks of established models of PD is essential for the neuroprotection that we have discussed in [22,83]. Previously, we have shown the Sigma1R-dependent neuroprotective effects of afobazole when the drug was administered over 14 days, starting on the day of 6-OHDA injection [22]. The use of selective Sigma1R ligands demonstrated the contribution of the chaperone to the effects of afobazole upon deferred administration (Figure 1, Figure 4 and Figure 5). As in the previous study, both afobazole and selective Sigma1R agonist PRE-084 (1.0 mg/kg i.p.) had similar normalizing effects on DA content and latency to fall in a rotarod test of ICR mice, which was blocked by the pre-administration of selective Sigma1R antagonist BD-1047 at a dose of 3.0 mg/kg i.p. [22].

According to Monville et al., performance in FSRR indicates overall motor impairment, whereas ARR is more sensitive to the severity of the nigrostriatal pathway lesion [91]. The effects of afobazole and PRE-084 administration in ARR and FSRR variations of the rotarod test on BD-1047 pretreatment point to the positive influence of Sigma1R activation on motor deficit and coordination impairment. The influence of striatal DA content on motor behavior was confirmed by a correlation analysis (Figure 6 and Figure 7). Interestingly, the difference in medians of latency to fall of afobazole and PRE-084 treated mice with 6-OHDA lesion and corresponding sham-operated animals in ARR was less than 8.5% (Figure 5, Appendix A). This observation may be the evidence of a restorative effect of these compounds on the nigrostriatal pathway.

In our study, the effect of Sigma1R ligands on DA content was accompanied by unidirectional changes in its metabolite concentrations. Along with the DA content increment, afobazole and PRE-084 upregulated DOPAC and HVA (Figure 2 and Figure 3). The restoration of DA metabolites is presumably predicted by the amplification of DA release within an established model of PD when up to 80% of SNc neurons are depleted [92]. On the other hand, Sigma1R activation facilitates DA release in the striatum [93,94]. These results are consistent with the ability of (+)-pentazocine to enhance, in a dose-dependent manner, the production of DOPAC and HVA in the rat striatum [95]. Experimental data gave evidence of the dependence of striatal dopaminergic system performance on the activation of Sigma1R. However, in microdialysis studies, the stimulation of DA release was observed at a dose of 10 mg/kg i.p. or s.c. of (+)-pentazocine [93,94], which are much higher than used in the present study’s Sigma1R agonists. In addition, the acute administration of Sigma1R selective agonist SA4503 does not affect dopamine content in the rat striatum [96], which is consistent with that of the sham-operated animals in our study. Taken together with data obtained in a rotarod test, an increase in intrastriatal DA content is compatible with the neurorestorative effect.

Interestingly, the administration of afobazole started at the day of 6-OHDA lesion did not cause a statistically significant increase in DA metabolites content in the lesioned striatum, which was discussed with regard to MAO-A inhibition by the drug [22]. The obtained results were accompanied by the lack of effect of MAO-A inhibitor clorgyline on DA and DOPAC content in a lesioned rat striatum two weeks after 6-OHDA injection [97].

Despite the restoration of DA, the deferred administration of afobazole did not cause a statistically significant increase of TH+ cells in the SNc (Table 1). These results are in line with the experimental data of Francardo et al., where the administration of PRE-084 (0.3 mg/kg s.c.) to C57Bl/6 mice over 35 days with a course starting seven days after surgery had no effect on TH+ count and locomotor activity [20]. At the same time, the administration of PRE-084 starting on the day of the lesion recovered the DA content in the striatum, increased the amount of TH+ cells in SNc, and facilitated mice locomotor activity in a battery of tests [20].

TH is the rate-limiting enzyme of DA synthesis, which determines the link between the number of TH+ neurons and DA content in PD modeling [22,98]. However, the loss of TH as a phenotypic trait of neurons in early stages of the lesion may be evidence of long-lasting atrophy as well as the downregulation of the TH enzyme in the surviving nigral DA neurons [99,100]. These observations accord with the wide variability in the number of TH+ neurons in the SNc of PD patients [101]. So, despite the absence of a significant effect on the number of SNc TH+ neurons, the restorative action of afobazole on DA and motor behavior may be conditioned by the induction of neuroprotective mechanisms.

We assume that caused by afobazole and PRE-084, the DA content increase and normalization of motor behavior, along with the absence of any influence on the number of TH+ neurons in SNc, are a result of the upregulation of TH activity. In ex vivo studies, selective Sigma1R agonist (+)-pentazocine increased TH activity in the rat striatum more than by 20%. The action of (+)-pentazocine was blocked by Sigma1R antagonist BMY-14802 [102]. In such a way, the restoration of intrastriatal DA content upon the deferred administration of Sigma1R agonists in the early stage PD model [6] is most likely caused through an increase in the TH activity of atrophic cells. On the other hand, it can be assumed that a further prolongation of afobazole course could result in a more pronounced restorative effect in TH+ cells, as it was demonstrated for PRE-084 by Francardo et al. [20].

Various molecular mechanisms may underlie the neurorestorative action of Sigma1R agonists. For example, the restorative effect of Sigma1R agonists on DA content in the striatum and locomotion depends on the upregulation of striatal brain-derived neurotrophic factor (BDNF) [20]. In vivo experiments demonstrated the ability of afobazole to increase the BDNF content in mice striatum [103]. It is possible that proteins of the BDNF signal cascades [104] are involved in the mechanism of Sigma1R-dependent action of afobazole. This assumption aligns with the ability of PRE-084 to enhance the phosphorylation of ERK 1/2, which was inhibited by the Sigma1R selective antagonist BD-1047 in primary neurons [105].

One of the major factors in the development of PD [106] and 6-OHDA [107] induced cell damage is microglia activation. Afobazole decreased the microglial cells migration induced by nucleoside triphosphates and amyloid Aβ_25–35_ in vitro [80,81]. In an in vivo model of ischemic stroke, afobazole prevents glial activation and death, while selective Sigma1R antagonists blocked the effect of the drug [77]. These stimuli of afobazole correspond to Sigma1R activation [58] and the neuroprotective effect of PRE-084 in the model of 6-OHDA-induced PD accompanied by the inhibition of microglia migration in SNc [20]. In addition to the effects common for Sigma1R agonists in the modeling of PD, afobazole may exert neuroprotective action through the activation of mechanisms identified in other experimental models utilizing selective Sigma1R antagonists. Afobazole prevents excessive Ca^2+^ influx [79,82] and decreases NO production [79]. The mentioned stimuli of afobazole align with the action of PRE-084 in vitro [57].

The present study indicates a contribution of the Sigma1R chaperone to the restorative effect of deferred afobazole administration on striatal DA content and motor behavior in a rotarod test of ICR mice in the model of PD induced by unilateral intrastriatal injection of 6-OHDA. The action of afobazole could be associated with the upregulation of TH activity and/or the activation of neuroprotective mechanisms. The obtained data allow for considering Sigma1R agonists as candidates for adjuvant therapy of PD.

## Figures and Tables

**Figure 1 ijms-21-07620-f001:**
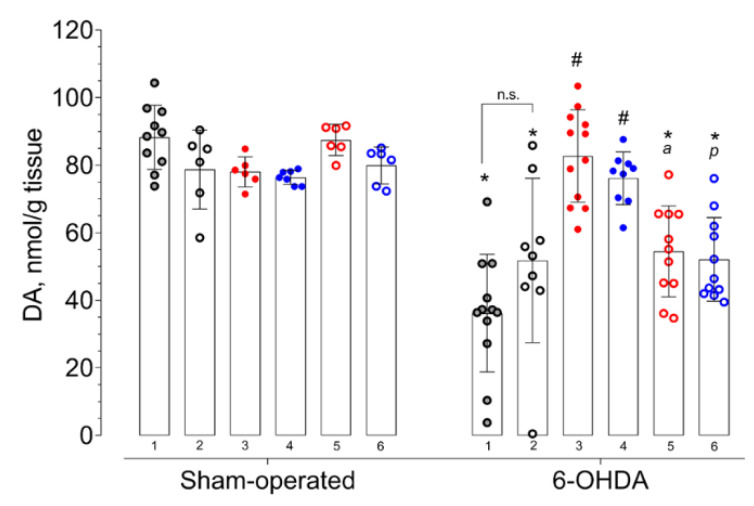
Influence of deferred administration of afobazole over 14 days on dopamine content in 6-hydroxidodamine (6-OHDA)-lesioned ICR mice striata. Data are presented as mean ± S.D. Sham-operated—ipsilateral striatum of sham-operated animals. 6-OHDA—ipsilateral striatum of animals with 6-OHDA lesion. Experimental groups divided by treatment, tissue harvesting, and last drug administration were spaced 1 h apart: **1** vehicle. **2** BD-1047 3.0 mg/kg. **3** afobazole 2.5 mg/kg. **4** PRE-084 1.0 mg/kg. **5** BD-1047 3.0 mg/kg and afobazole 2.5 mg/kg. **6** BD-1047 3.0 mg/kg and PRE-084 1.0 mg/kg. Two-way ANOVA, Tukey multiple comparison test: A significant difference (*p* < 0.01) was observed in 6-OHDA-lesioned mice within all experimental groups except for afobazole (3) and PRE-084 (4). Statistical significance: *****
*p* < 0.01 vs. sham-operated vehicle-treated mice. **#**
*p* < 0.01 vs. 6-OHDA-lesioned vehicle-treated mice. **a**
*p* < 0.01 vs. 6-OHDA-lesioned afobazole-treated mice. **p**
*p* < 0.05 vs. 6-OHDA-lesioned PRE-084-treated mice. **n.s.**—no statistical significance between compared groups. Paired *t*-test: A significant difference (*p* < 0.001) was observed between the contralateral and ipsilateral striata of 6-OHDA-lesioned mice for all groups.

**Figure 2 ijms-21-07620-f002:**
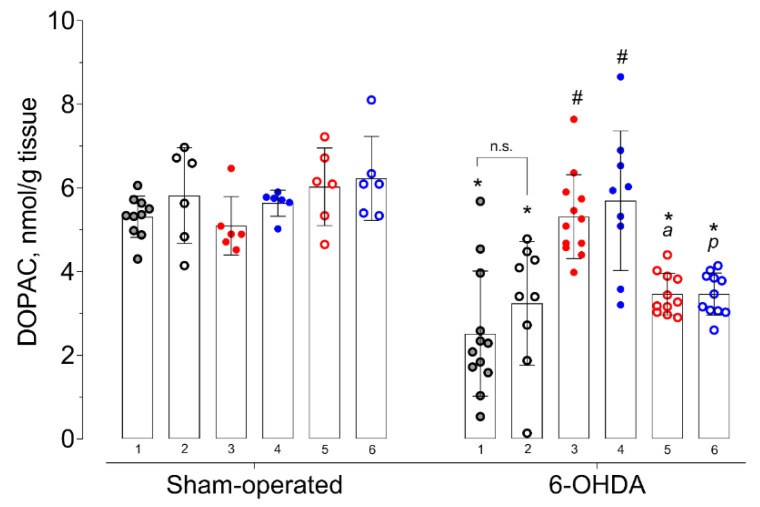
Influence of deferred administration of afobazole over 14 days on DOPAC content in 6-OHDA-lesioned ICR mice striata. Data are presented as mean ± S.D. Sham-operated—ipsilateral striatum of sham-operated animals. 6-OHDA—ipsilateral striatum of animals with 6-OHDA lesion. Experimental groups were divided by treatment, tissue harvesting, and last drug administration and spaced 1 h apart: **1** vehicle. **2** BD-1047 3.0 mg/kg. **3** afobazole2.5 mg/kg. **4** PRE-084 1.0 mg/kg. **5** BD-1047 3.0 mg/kg and afobazole 2.5 mg/kg. **6** BD-1047 3.0 mg/kg and PRE-084 1.0 mg/kg. Two-way ANOVA, Tukey multiple comparison test: A significant difference (*p* < 0.05) was observed within all experimental groups except for afobazole (3) and PRE-084 (4). Statistical significance: *****
*p* < 0.01 vs. sham-operated vehicle-treated mice. **#**
*p* < 0.01 vs. 6-OHDA-lesioned vehicle-treated mice. **a**
*p* < 0.01 vs. 6-OHDA-lesioned afobazole-treated mice. **p**
*p* < 0.05 vs. 6-OHDA-lesioned PRE-084-treated mice. **n.s.**—no statistical significance between compared groups. Paired *t*-test: A significant difference (*p* < 0.001) was observed between the contralateral and ipsilateral striata of 6-OHDA-lesioned mice for all groups except for afobazole and PRE-084.

**Figure 3 ijms-21-07620-f003:**
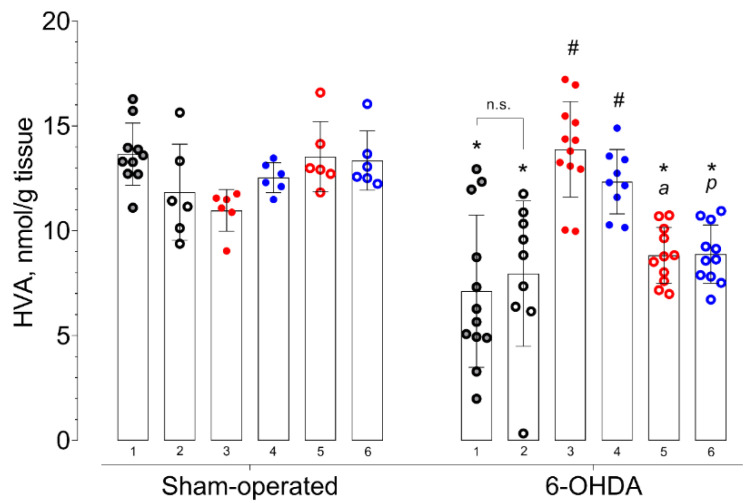
Influence of deferred administration of afobazole over 14 days on HVA content in 6-OHDA-lesioned ICR mice striata. Data are presented as mean ± S.D. Sham-operated—ipsilateral striatum of sham-operated animals. 6-OHDA—ipsilateral striatum of animals with 6-OHDA lesion. Experimental groups were divided by treatment, tissue harvesting and last drug administration were 1-h spaced apart: **1** vehicle. **2** BD-1047 3.0 mg/kg. **3** afobazole 2.5 mg/kg. **4** PRE-084 1.0 mg/kg. **5** BD-1047 3.0 mg/kg and afobazole 2.5 mg/kg. **6** BD-1047 3.0 mg/kg and PRE-084 1.0 mg/kg. Two-way ANOVA, Tukey multiple comparison test: A significant difference (*p* < 0.05) was observed within all experimental groups except for afobazole (3) and PRE-084 (4). Statistical significance: *****
*p* < 0.01 vs. sham-operated vehicle-treated mice. **#**
*p* < 0.01 vs. 6-OHDA-lesioned vehicle-treated mice. **a**
*p* < 0.01 vs. 6-OHDA-lesioned afobazole-treated mice. **p**
*p* < 0.05 vs. 6-OHDA-lesioned PRE-084-treated mice. **n.s.**—no statistical significance between compared groups.

**Figure 4 ijms-21-07620-f004:**
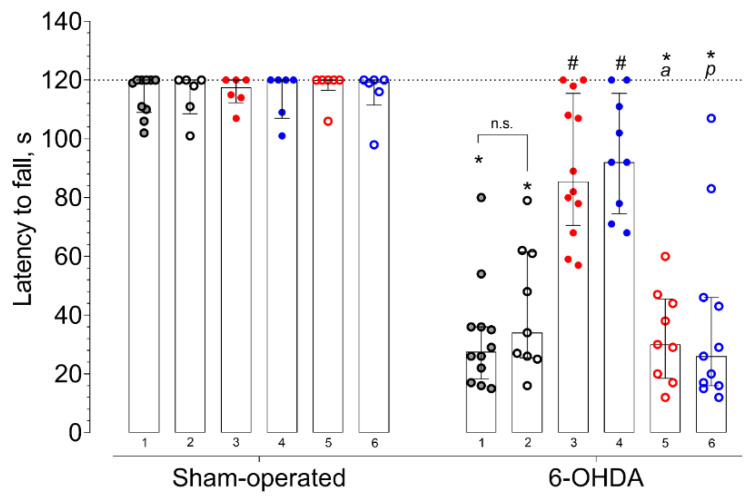
The influence of deferred administration of afobazole over 14 days on latency to fall during fixed-speed rotarod test (FSRR) in 6-OHDA-lesioned and sham-operated ICR mice. Data are presented as median with interquartile range. Sham-operated—sham-operated animals. 6-OHDA—animals with 6-OHDA lesion. Experimental groups were divided by treatment, drugs were administered 20 h prior to the test: **1** vehicle. **2** BD-1047 3.0 mg/kg. **3** afobazole 2.5 mg/kg. **4** PRE-084 1.0 mg/kg. **5** BD-1047 3.0 mg/kg and afobazole 2.5 mg/kg. **6** BD-1047 3.0 mg/kg and PRE-084 1.0 mg/kg. Kruskal‒Wallis test, Dunn’s multiple comparison test: A significant difference (*p* < 0.05) was observed within all experimental groups except for afobazole (3) and PRE-084 (4). Statistical significance: *****
*p* < 0.01 vs. sham-operated vehicle-treated mice. **#**
*p* < 0.01 vs. 6-OHDA-lesioned vehicle-treated mice. **a**
*p* < 0.01 vs. 6-OHDA-lesioned afobazole-treated mice. **p**
*p* < 0.05 vs. 6-OHDA-lesioned PRE-084-treated mice. **n.s.**—no statistical significance between compared groups.

**Figure 5 ijms-21-07620-f005:**
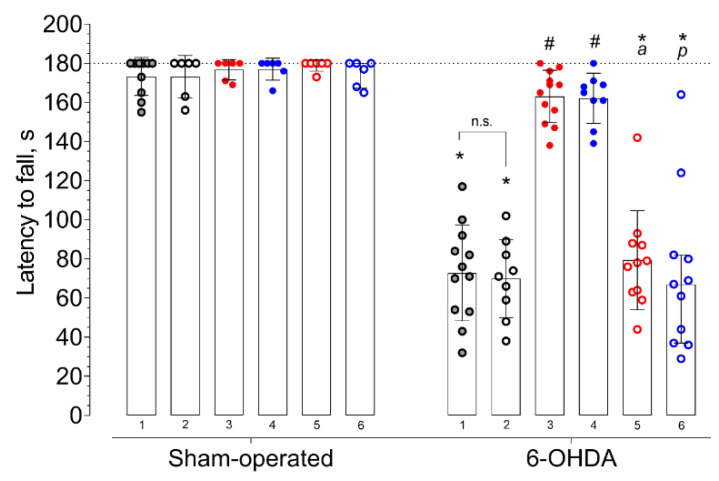
The influence of deferred administration of afobazole over 14 days on latency to fall during accelerated rotarod test (ARR) in 6-OHDA-lesioned and sham-operated ICR mice. Data are presented as median with interquartile range. Sham-operated—sham-operated animals. 6-OHDA—animals with 6-OHDA lesion. Experimental groups were divided by treatment, drugs were administered 20 h prior to the test: **1** vehicle. **2** BD-1047 3.0 mg/kg. **3** afobazole 2.5 mg/kg. **4** PRE-084 1.0 mg/kg. **5** BD-1047 3.0 mg/kg and afobazole 2.5 mg/kg. **6** BD-1047 3.0 mg/kg and PRE-084 1.0 mg/kg. Kruskal‒Wallis test, Dunn’s multiple comparison test: A significant difference (*p* < 0.05) was observed within all experimental groups except for afobazole (3) and PRE-084 (4). Statistical significance: *****
*p* < 0.01 vs. sham-operated vehicle-treated mice. **#**
*p* < 0.05 vs. 6-OHDA-lesioned vehicle-treated mice. **a**
*p* < 0.01 vs. 6-OHDA-lesioned afobazole-treated mice. **p**
*p* < 0.05 vs. 6-OHDA-lesioned PRE-084-treated mice. **n.s.**—no statistical significance between compared groups.

**Figure 6 ijms-21-07620-f006:**
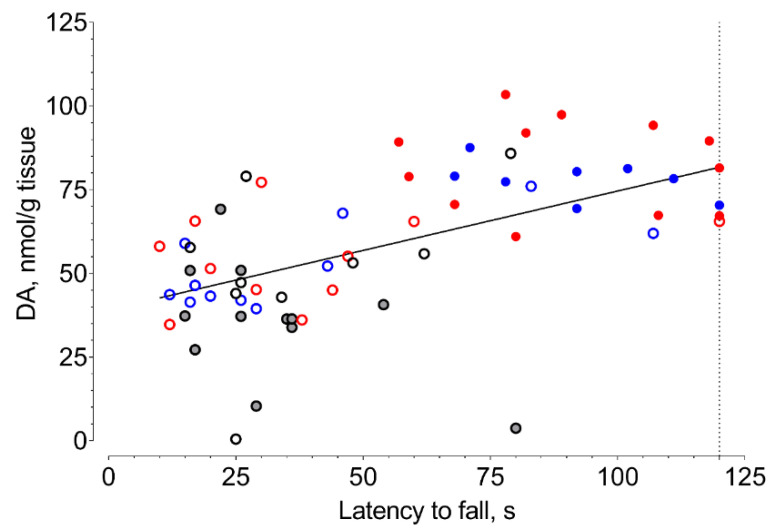
Correlation between striatal dopamine content of 6-OHDA-lesioned mice and their latency to fall in a fixed-speed rotarod test (FSRR). **Gray dot**—vehicle. **Black circle**—BD-1047 3.0 mg/kg. **Red dot**—afobazole 2.5 mg/kg. **Blue dot**—PRE-084 1.0 mg/kg. **Red circle**—BD-1047 3.0 mg/kg and afobazole 2.5 mg/kg. **Blue circle**—BD-1047 3.0 mg/kg and PRE-084 1.0 mg/kg. R = 0.6, *p* < 0.0001; line of best fit equation DA = 0.3518*X + 39.11.

**Figure 7 ijms-21-07620-f007:**
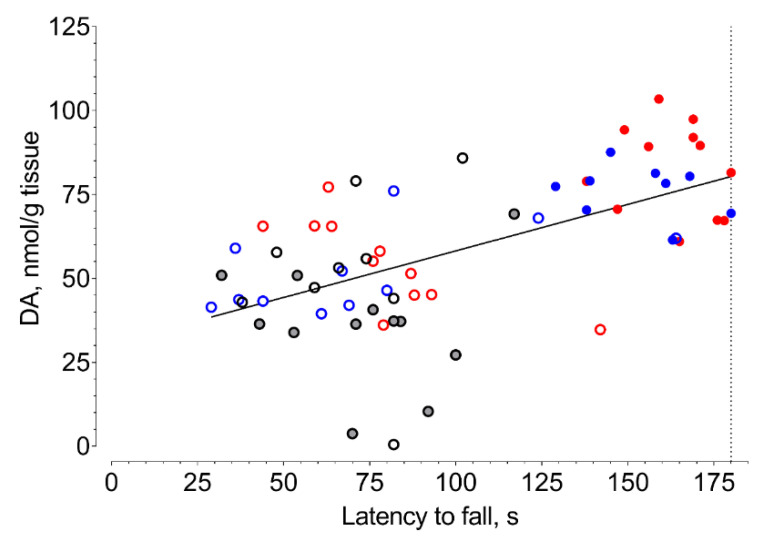
Correlation between striatal dopamine content of 6-OHDA-lesioned mice and their latency to fall in accelerated rotarod test (ARR). **Gray dot**—vehicle. **Black circle**—BD-1047 3.0 mg/kg. **Red dot**—afobazole 2.5 mg/kg. **Blue dot**—PRE-084 1.0 mg/kg. **Red circle**—BD-1047 3.0 mg/kg and afobazole 2.5 mg/kg. **Blue circle**—BD-1047 3.0 mg/kg and PRE-084 1.0 mg/kg. R = 0.6, *p* < 0.0001; line of best fit equation DA = 0.2822*X + 29.05.

**Table 1 ijms-21-07620-t001:** The influence of afobazole on the tyrosine hydroxylase-positive (TH+) cell count.

Experimental Groups	TH+ Neurons (Neurons/Slide)	Relative TH+ (%)
Sham + Vehicle	93.3 (83.75–108.0)	94.86 (89.24–104.1)
6-OHDA + Vehicle	60.2 (16.55–65.5) * *p* = 0.027	61.6 (38.65–84.96) * *p* = 0.022
6-OHDA + Afobazole	61.5 (35.4–72.5) * *p* = 0.04	82.12 (57.81–87.07)

Data are presented as median (q25–75). The number of animals in each experimental group was five. *—statistical significance vs. vehicle-treated sham-operated animals (Kruskal‒Wallis test, Dunn’s post hoc test).

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
