# Peer review of "Deferred Administration of Afobazole Induces Sigma1R-Dependent Restoration of Striatal Dopamine Content in a Mouse Model of Parkinson’s Disease"

_ijms, 2020, doi:10.3390/ijms21207620_

Round 1
Reviewer 1 Report
This paper shows significant functional effects of delayed-start afobazole treatment in an intrastriatal 6-OHDA model of PD in mice; the observed effects are mediated by the Sigma-1 receptor as they are blocked by a Sigma-1 antagonist. This is a well-written and competently discussed study, with clearly presented and interesting results. I have two reservations, however:
1) The same authors published a very similar study in Sci Rep 2019 (PMID 31745133). The animal model, drug treatments, and functional-histological-biochemical endpoints appear to be the same as in the present study, with the only difference that, in the present study, the drug treatments started 14 days post-lesion. Having said that, I leave it to the Editors to decide whether there is sufficient novelty in the present submission to qualify for a publication in IJMS.
2) As the authors correctly point out, there is discrepancy between the lack of a neuroprotective effect by afobazole on nigral dopamine neurons and its remarkable restoration of striatal DA levels as well as behaviour. In light of previous reports that this drug and other sigma-1 agonists may acutely stimulate DA release, the authors need to clearly specify (already in the Results section) what was the interval between last drug administration and (a) the behavioural tests reported in Figs 4 and 5; (b) the harvesting of striatal tissue for biochemical determinations of DA and its metabolites.
3) Related to the same issue, the Discussion needs to be clearer on whether the results here reported reflect a symptomatic improvement or a true neurorestorative effect (which may have potential impact on the disease course in human PD).
Author Response
Response to Reviewer 1 Comments
We thank the Reviewer for deep and detailed analysis of our manuscript. We appreciate your conclusions and comments.
Point 1: The same authors published a very similar study in Sci Rep 2019 (PMID 31745133). The animal model, drug treatments, and functional-histological-biochemical endpoints appear to be the same as in the present study, with the only difference that, in the present study, the drug treatments started 14 days post-lesion. Having said that, I leave it to the Editors to decide whether there is sufficient novelty in the present submission to qualify for a publication in IJMS.
Response 1: Dear Reviewer points on high level of similarities between the current study and our previous work published in Scientific Reports. However, we are certain that the investigation of pharmacological regulation of potential protein targets within the frameworks of established pathology and, therefore, administration of drugs applied at deferred time point is crucial for the development of medications for the treatment of neurodegenerative disorders. In these experimental conditions, pathological changes typical for Parkinson’s disease are formed two weeks after the lesion, which we have demonstrated in [1] and discussed in [2]. Also, we would like to point the Reviewer’s attention to the additional strength of the present study, particularly the presence of various control groups essential for comprehension of obtained results. Specifically, sham-operated animals treated with compounds of interest and its combinations as well as 6-hydroxidopamine-lesioned mice treated with Sigma1R antagonist BD-1047. Besides, as we indicate in “Materials and Methods” (lines 376 – 380) and “Discussion” sections (lines 251 - 254) to study motor behavior and coordination rotarod test was utilized in two modifications, as follows fixed-speed (FSRR) and accelerated (ARR) rotarod. FSRR test allowed us to directly compare the results of present study with previously published [1]. At the same time, more sensitive to the severity of the nigrostriatal lesion ARR test [3] was included for the precise analysis of compounds effects.
Point 2: As the authors correctly point out, there is discrepancy between the lack of a neuroprotective effect by afobazole on nigral dopamine neurons and its remarkable restoration of striatal DA levels as well as behaviour. In light of previous reports that this drug and other sigma-1 agonists may acutely stimulate DA release, the authors need to clearly specify (already in the Results section) what was the interval between last drug administration and (a) the behavioural tests reported in Figs 4 and 5; (b) the harvesting of striatal tissue for biochemical determinations of DA and its metabolites.
Response 2: We agree with the Reviewer comment on the ability of Sigma1R agonists to stimulate dopamine release at acute administration that may affect rotarod performance. In our research, drugs were administered 20 hours prior to rotarod test that in our opinion excludes the influence of acute administration on motor behavior. The last drug administration was performed 1 hour prior to tissue harvesting. Microdialysis studies demonstrate that in the pointed time interval striatal dopamine release is stimulated by prototypic Sigma1R agonist (+)-pentazocine at doses of 10 mg/kg i.p. [4] and 10 mg/kg s.c. [5], that are much higher than the doses of Sigma1R agonists used in our study. Moreover, acute administration of Sigma1R selective agonist SA4503 does not affect dopamine content in the rat striatum [6], which is consistent with that of sham-operated animals in our study. Following the Reviewer’s suggestions we have modified “Results” (lines 134 – 135; 163 – 164; 177 – 178; 212 – 213; 224 – 225 of the revised manuscript) and “Materials and Methods” (lines 319 – 320; 325 – 326 of the revised manuscript) sections.
Point 3: Related to the same issue, the Discussion needs to be clearer on whether the results here reported reflect a symptomatic improvement or a true neurorestorative effect (which may have potential impact on the disease course in human PD).
Response 3: In our opinion, facilitated performance of 6-hydroxidopamine-lesioned mice in ARR test along with known ability of Sigma1R agonists to stimulate dopamine release, but only in the high doses, makes us to consider increased intrastriatal dopamine as a sign of neurorestoration. According to the Reviewer’s comments, we have modified “Discussion” section (lines 382 – 383; 391 – 392; 402 – 404; 408 – 413; 434 – 442 of revised manuscript).
- Voronin, M.V.; Kadnikov, I.A.; Voronkov, D.N.; Seredenin, S.B. Chaperone Sigma1R mediates the neuroprotective action of afobazole in the 6-OHDA model of Parkinson's disease. Sci Rep 2019, 9, 17020, doi:10.1038/s41598-019-53413-w.
- Voronin, M.V.; Kadnikov, I.A.; Seredenin, S.B. Afobazole Restores the Dopamine Level in a 6-Hydroxydopamine Model of Parkinson’s Disease. Neurochemical Journal 2019, 13, 49-56, doi:10.1134/S1819712419010185.
- Monville, C.; Torres, E.M.; Dunnett, S.B. Comparison of incremental and accelerating protocols of the rotarod test for the assessment of motor deficits in the 6-OHDA model. J Neurosci Methods 2006, 158, 219-223, doi:10.1016/j.jneumeth.2006.06.001.
- Patrick, S.L.; Walker, J.M.; Perkel, J.M.; Lockwood, M.; Patrick, R.L. Increases in rat striatal extracellular dopamine and vacuous chewing produced by two sigma receptor ligands. Eur J Pharmacol 1993, 231, 243-249, doi:10.1016/0014-2999(93)90456-r.
- Gudelsky, G.A. Effects of sigma receptor ligands on the extracellular concentration of dopamine in the striatum and prefrontal cortex of the rat. Eur J Pharmacol 1995, 286, 223-228, doi:10.1016/0014-2999(95)00415-8.
- Kobayashi, T.; Matsuno, K.; Murai, M.; Mita, S. Sigma 1 receptor subtype is involved in the facilitation of cortical dopaminergic transmission in the rat brain. Neurochem Res 1997, 22, 1105-1109, doi:10.1023/a:1027361101419.
Reviewer 2 Report
The study is a nice follow up of the previous work by the authors and adds important information. I have only a few minor comments.
- Authors should provide a better explanation why despite the restoration of DA, the deferred administration of afobazole did not cause a statistically significant increase of TH+ cells in the SNc.
- Authors should include a rationale for performing two versions (fixed-speed and accelerated) of rotarod tests.
- Though it is not significant, the DA content in (6-OHDA+BD-1047) treated mice (group 2) is higher than (6-OHDA+vehicle) treated mice (group 1), Fig. 1. Authors should include some possible explanation for this.
Author Response
Response to Reviewer 2 Comments
We thank the Reviewer for deep analysis of our manuscript and detailed comments.
Point 1: Authors should provide a better explanation why despite the restoration of DA, the deferred administration of afobazole did not cause a statistically significant increase of TH+ cells in the SNc.
Response 1: We assume that restoration of dopamine observed in afobazole-treated mice with 6-OHDA lesion could be caused by increase of TH activity in injured SNc cells. Meanwhile, more extended course of afobazole could result in more pronounced effect on viability of TH+ cells as it was demonstrated in [1] for PRE-084. We have modified “Discussion” section (lines 434 – 442) according to the Reviewer suggestions.
Point 2: Authors should include a rationale for performing two versions (fixed-speed and accelerated) of rotarod tests.
Response 2: We thank the Reviewer for this comment. Designing our experiments we took in consider previously published work by Monville et al., who demonstrated that performance in FSRR indicates overall motor impairment, whereas ARR is more sensitive to the severity of the nigrostriatal pathway lesion in rat 6-OHDA induced model of Parkinson’s disease [2]. In our point of view, the use of two variations of the rotrod test allows to evaluate the influence of Sigma1R ligands in mice in this model more precisely. According to the Reviewer’s comments, we made appropriate changes in “Materials and Methods” (lines 309 – 311 of the revised manuscript) and “Discussion” (lines 391 – 392 of the revised manuscript) sections.
Point 3: Though it is not significant, the DA content in (6-OHDA+BD-1047) treated mice (group 2) is higher than (6-OHDA+vehicle) treated mice (group 1), Fig. 1. Authors should include some possible explanation for this.
Response 3: Regarding interesting comment of the Reviewer in accordance with Garcés-Ramírez et al. BD-1047 is micromolar dopamine transporter (DAT) ligand [3] and the meaning of such interaction could be revealed in [4].
- Francardo, V.; Bez, F.; Wieloch, T.; Nissbrandt, H.; Ruscher, K.; Cenci, M.A. Pharmacological stimulation of sigma-1 receptors has neurorestorative effects in experimental parkinsonism. Brain 2014, 137, 1998-2014, doi:10.1093/brain/awu107.
- Monville, C.; Torres, E.M.; Dunnett, S.B. Comparison of incremental and accelerating protocols of the rotarod test for the assessment of motor deficits in the 6-OHDA model. J Neurosci Methods 2006, 158, 219-223, doi:10.1016/j.jneumeth.2006.06.001.
- Garces-Ramirez, L.; Green, J.L.; Hiranita, T.; Kopajtic, T.A.; Mereu, M.; Thomas, A.M.; Mesangeau, C.; Narayanan, S.; McCurdy, C.R.; Katz, J.L., et al. Sigma receptor agonists: receptor binding and effects on mesolimbic dopamine neurotransmission assessed by microdialysis. Biol Psychiatry 2011, 69, 208-217, doi:10.1016/j.biopsych.2010.07.026.
- Sambo, D.O.; Lebowitz, J.J.; Khoshbouei, H. The sigma-1 receptor as a regulator of dopamine neurotransmission: A potential therapeutic target for methamphetamine addiction. Pharmacol Ther 2018, 186, 152-167, doi:10.1016/j.pharmthera.2018.01.009.